# Occupational Lyme Disease: A Systematic Review and Meta-Analysis

**DOI:** 10.3390/diagnostics12020296

**Published:** 2022-01-25

**Authors:** Nicola Magnavita, Ilaria Capitanelli, Olayinka Ilesanmi, Francesco Chirico

**Affiliations:** 1Post-Graduate School of Occupational Medicine, Università Cattolica del Sacro Cuore, 00168 Rome, Italy; medlavchirico@gmail.com; 2Department of Science of Woman, Child & Public Health, Fondazione Policlinico Agostino Gemelli IRCSS, 00168 Rome, Italy; 3Prevention Service in the Workplace (SPRESAL), Local Health Unit Roma 4, 00053 Civitavecchia, Italy; ilaria.capitanelli@yahoo.it; 4Department of Community Medicine, College of Medicine, University of Ibadan, Ibadan 200281, Nigeria; ileolasteve@yahoo.co.uk; 5Health Service Department, Italian State Police, Ministry of the Interior, 20123 Milan, Italy

**Keywords:** public health, tick-borne diseases, infectious disease, occupational health, outdoor workers, seroprevalence, diagnosis

## Abstract

Lyme disease (LD) can have significant consequences for the health of workers. The frequency of infection can be estimated by using prevalence and incidence data on antibodies against Borrelia Burgdoferi (BB). A systematic search of studies published in English between 2002 and 2021 and a meta-analysis were conducted in PubMed/Medline, Web of Science, Scopus, and Google Scholar databases. Out of a total of 1125 studies retrieved, 35 articles were included in the systematic review. Overall, in these studies, outdoor workers showed a 20.5% BB seroprevalence rate. Meta-analysis, performed on 15 studies (3932 subjects), revealed a significantly increased risk in outdoor activities (OR 1.93 95%CI 1.15–3.23), with medium-level heterogeneity (I^2^ = 69.2%), and non-significant publication bias. The estimated OR in forestry and agricultural workers was 2.36 (CI95% 1.28; 4.34) in comparison with the controls, while a non-significant increase in risk (OR = 1.05, CI95% 0.28; 3.88) was found in the remaining categories of workers (veterinarians, animal breeders, soldiers). The estimated pooled risk was significantly higher in the studies published until 2010 (OR 3.03 95%CI 1.39–6.61), while in more recent studies the odds became non-significant (OR 1.08 95% CI 0.63–1.85). The promotion of awareness campaigns targeting outdoor workers in endemic areas, and the implementation of local programs aimed at controlling range expansion of vectors, are key strategies for protecting workers.

## 1. Introduction

Lyme disease (LD) is an infectious disease caused by Borrelia burgdorferi (BB), a tick-borne bacterium, commonly found in animals such as mice and deer. Ixodes ticks can become infected with the bacteria when they bite an infected animal; they can then pass it on to a person through a subsequent bite. LD is among the most frequently diagnosed zoonotic tick-borne diseases worldwide [1]; approximately 300,000 people in the USA and 65,000 in Europe [2] manifest clinical symptoms of LD annually. These symptoms include a bullseye-shaped rash (Erythema migrans), fever, chills, headache, fatigue, muscle and joint pain, and swollen lymph nodes [3,4]. If LD is diagnosed and treated with antibiotics, most cases do not go beyond a skin rash. However, if it is left undetected, various long-lasting neurologic, cardiovascular, and musculoskeletal long-term consequences may occur [5]. The late disseminated stage manifests as acrodermatitis atrophicans, Lyme arthritis, and neurological symptoms can be seriously debilitating [6]. The immune-mediated and metabolic changes can induce multiple debilitating symptoms and alter neural circuits. Neurological manifestations, termed Lyme neuroborreliosis, occur in approximately 10% of patients with LD [7,8]. A subset of patients may report persistent symptoms, including severe fatigue, anxiety and depression [9]. A small percentage (~10%) of patients may go on to develop a poorly defined fibromyalgia-like illness, post-treatment Lyme disease symptoms [10], whose characterization is still controversial. Hospitalized patients have an increased risk of mental disorders, affective disorders, suicide attempts and suicide deaths compared with other patients without LD [11]. Results of indirect estimates reveal that there may be more than 1200 LD-associated suicides in the US per year [12]. 

LD diagnosis is challenging due to the varied clinical manifestations it may present [6] and is supported by serologic testing using a 2-step process [13,14]. Current recommendations based on the 1995 guidelines of the Centers for Disease Control and Prevention include using a sensitive enzyme immunoassay or immunofluorescence assay, such as enzyme-linked immunosorbent assay (ELISA) or immunofluorescent assay (IFA), followed by a western immunoblot assay for specimens yielding positive or equivocal results [15]. In 2019, the Food and Drug Administration (FDA) cleared several LD serologic assays with new indications for use, allowing for an enzyme immunoassay rather than western immunoblot assay as the second test in a Lyme disease testing algorithm [16]. The two-step approach is intentionally conservative to exclude false positive results; consequently, it is prone to false negative results that lead to underestimation of the number of people with LD. In Canada, a 10 to 20-fold underestimate has been calculated [17]. The Western blot/immunoblot assay confirms a percentage between 30% and 70% of positivity at first assay [18,19] However, some epidemiological studies, especially in past years, have used a single ELISA test approach for detecting BB infection.

LD has dramatically expanded its geographic range over the last two decades [20]. Global warming, deforestation, and changes in precipitation [21], as well as other environmental conditions (habitat, climate, and dust and gas pollution) prevailing within urban heat islands have probably exerted an impact on tick abundance and activity, thus increasing the prevalence of LD over extensive areas [22]. Vegetation type and distribution (canopy, understory, and ground cover), human behavior [23], and seasons [24] have been found to be associated with an increased risk of tick bites [24]. 

According to occupational health experts, the potential occupational risk of LD among outdoor workers is an important issue, especially for those working in LD-endemic areas [25,26,27]. Forestry workers, who carry out their activity in areas of wild vegetation, may be at a high risk of LD [28]. Other categories at risk for LD include outdoor workers employed in uncultivated areas, military personnel based in endemic areas or working in wooded areas, and veterinarians who have had contact with tick-bearing animals [29,30,31]. 

The main aim of this paper was to systematically review the occurrence of LD in workers, evaluate the prevalence of antibodies against BB, and conduct a meta-analysis to estimate the relative risk for outdoor workers compared to controls. A secondary aim of this study was to identify LD risk factors in occupational cohorts in order to obtain elements for evidence-based prevention policies.

## 2. Materials and Methods

We conducted this systematic review and meta-analysis using a pre-established protocol, which was registered on the international prospective registry for systematic review protocols PROSPERO (registration number: CRD42021252608). 

### 2.1. Search Strategy

We followed the Preferred Reporting Items for Systematic Reviews and Meta-Analyses (PRISMA), and the Cochrane criteria [32]. We searched PubMed/Medline, Web of Science, Scopus, and Google Scholar databases in English for articles (published from 1 January 2002 to 31 March 2021) that complied with the following criteria:

### 2.2. Inclusion Criteria

Study design: published, peer-reviewed randomized control trials (RCTs), case series and observational studies. Population: workers. Intervention: exposure to BB. Comparison: workers not exposed to BB. Outcome: seroprevalence of LD. Settings: occupational settings globally. 

### 2.3. Exclusion Criteria

All of the theoretical models included: editorials, reviews, guidelines, and public press articles were excluded. We also excluded case series without seroprevalence data and research on non-occupational cohorts (including subjects whose exposure to BB was exclusively related to outdoor recreational activities such as hunting, fishing, gardening, etc.). Research on vaccines against LD, genetic or diagnostic aspects of LD, neurological sequelae, or knowledge and literacy were also excluded. 

The following Medical Subject Heading (MeSH) terms were used: occupational groups; occupational medicine; industry; occupational diseases; disease; employment; occupational health; occupations; workplace; occupational exposure; workload and work. When building the search syntax, for prompt identification of studies conducted in an occupational setting, we referred to the strings Mattioli et al. [33] specifically developed for this purpose and used the ‘most sensitive search strategy’ (occupational diseases [MH] OR occupational exposure [MH] OR occupational exposure* [TW] OR ‘occupational health’ OR ‘occupational medicine’ OR work-related OR working environment [TW] OR at work [TW] OR work environment [TW] OR occupations [MH] OR work [MH] OR workplace* [TW] OR workload OR occupation* OR worker* OR work place* [TW] OR work site* [TW] OR job* [TW] OR occupational groups [MH] OR employment OR worksite* OR industry) AND Lyme Borreliosis OR Lyme disease). A similar string has been successfully used in a previous systematic review on emerging zoonotic viral infections of occupational health importance [34]. In addition, reference lists of included studies were searched for relevant publications that met the inclusion criteria. 

### 2.4. Data Extraction and Assessment Bias

The studies were grouped according to the following categories: (1) Occupational category of workers; (2) Geographic area of the study; (3) Time of the study; (4) Serological investigations performed. The predictable heterogeneity across our study was investigated by referring to the above categories. After an initial screening of the title and abstract, we read the full text of eligible studies. Two independent authors (FC and IC) performed study selection, and differences were resolved by a third author (NM). Data were collected from each relevant study. Extracted information included: (i) Source (first author and year of publication); (ii) General study details (citation, study design and year of publication); (iii) Setting (country/region considered, study population vs comparison group, and type of employment); (iv) Exposure measurement details (methodology concerning the diagnostic tools used, length of service, and number of tick-borne bites, if available); (v) Main findings of the study.

### 2.5. Meta-Analysis

Studies involving a control group were selected for meta-analysis. The meta-analysis was performed with the Meta-Essentials package, version 1.5 [35,36]. The pooled odds ratio (OR) of the studies was obtained by using the random effects model [37]. We examined the existence of heterogeneity among primary studies and analyzed the variance in the results of different studies. The consistency of the results was tested by the heterogeneity indicator (I square—I^2^-statistic), with I^2^ values of 25%, 50%, and 75% corresponding to a small, medium, and large degree of heterogeneity, respectively. Furthermore, the publication bias of the five effect sizes was tested by visual inspection. An asymmetric shape in the funnel plots implied the existence of publication bias [38]. 

### 2.6. Quality Appraisal

The quality of cohort and case-control studies was assessed by means of the Newcastle-Ottawa scale (NOS) that evaluates with a maximum score of 9 points, selection, comparability, and exposure criteria [39,40]. The quality of the other studies was evaluated using the adapted Newcastle-Ottawa Quality Assessment Scale (NOS-A) for case-control/cross-sectional studies [41]. This scale gives a maximum score of 10 points.

## 3. Results

Research on databases resulted in a total of 1125 studies, while 15 articles were retrieved from other sources. After a removal of the duplicates and studies that failed to meet the eligibility criteria, 135 full-text articles were assessed. A total of 5 literature reviews were excluded, together with 27 studies concerning non-occupational or mixed cohorts, a study conducted on an undefined cohort of workers, 9 case series without seroprevalence data, 12 studies on genetic or clinical aspects of the disease, 19 studies written in non-English languages, 9 studies on public health strategies for disease control, 12 studies on awareness of and attitudes towards tick-borne diseases, and 3 studies on infections accompanying other tick-borne diseases. Three studies were unavailable. 

In the end, the systematic review included 35 studies, 15 of which contained prevalence data in exposed workers and controls and were included in the quantitative meta-analysis (Figure 1). 

Of the 35 selected studies (Table 1), 25 investigated forestry workers and/or farmers, while the others studied mixed cohorts of forestry workers and soldiers, police officers, hunters, gardeners, or veterinarians. A total of 32 studies had a cross-sectional design, and 3 had a prospective design. Moreover, 31 studies were from Europe (8 from Poland, 5 from Italy, 4 from Turkey; the remainder were from Germany, Belgium, Finland, Slovakia, Slovenia, Hungary, Serbia, France, and Spain); 2 from North and Central America, and 2 from Asia. Eleven studies were conducted in the first decade (2002–2011), while the remainder were carried out in the second decade of observation. 

Researchers usually employed enzyme-linked immunosorbent assay (ELISA) or immunofluorescent assay (IFA) tests to determine the presence of IgG/IgM antibodies; eighteen studies also confirmed diagnosis with western blot methods (WB), according to CDC guidelines. In 23 studies, the serological investigation was supplemented by a clinical-epidemiological interview with a questionnaire, while in 5 studies, workers underwent a physical examination. In the few studies that provided this data, the proportion of seropositive workers reporting symptoms of LD, in particular erythema migrans, ranged considerably from low (6.5% [47],10.1% [55], 10.2% [55], 13.4% [43]) and medium (28.1% [69]) to high percentages (63.6% [44]). Two studies found no clinical evidence of Lyme disease symptomatology in workers who tested positive [52], or in those who seroconverted [46]. In the only study that reported these data, the workers with a positive confirmation test (WB) were all symptomatic [50].

### 3.1. Quality of Studies

According to our evaluation, the studies retrieved had, on average, a moderate to low quality score (ranging from 4 to 5 on the 9-point NOS scale, and from 2 to 7 on the 10-point NOS-A scale). The most negative aspect of the studies was the sampling process. Since 20 (55%) studies were simply aimed at studying the prevalence of BB infection in specific occupational groups, the authors had enrolled small convenience samples of outdoor workers—in some cases without including a control group. In addition, few studies indicated the response rate, and none reported the characteristics of the non-responders group. When a control group was available, it generally consisted of unmatched individuals (e.g., healthy blood donors with no information on their exposure). In other cases, the researchers used workers of the same company who performed mainly office duties as controls, or compared a category considered to be at higher risk with one at lower risk. 

### 3.2. Exposure

For all of the workers, the source of infection was the occurrence of tick bites during outdoor work activities in areas infested by infected vectors. This hypothesis was indirectly confirmed by the higher seroprevalence of anti-BB antibodies in workers reporting a high number of tick bites [58,67,69,73], and was demonstrated by identifying BB genetic material in ticks [42,47]. 

### 3.3. Prevalence of BB Infection

Subjects employed in agriculture and forestry were the occupational groups most frequently affected by BB infection, with 4428 positive individuals out of 21,546 workers in 35 studies (20.5% BB seroprevalence). A subgroup analysis showed that forestry workers had a higher BB antibody rate (25%) than farmers (14%). Veterinarians and animal breeders were also assumed to be at risk of contracting LD during their occupational activities (e.g., breeding and visiting farms and pastures that could host infected ticks). However, BB prevalence in these categories was similar or slightly higher than that of the general population [61], and much lower than that of farmers [48]. 

The annual incidence of LD was evaluated only in the three prospective studies, where it was zero [53], 4.4% [46] and 9% [48]. The relative risk was not significant in these studies.

### 3.4. Risk Factors

Frequent or permanent contact with forest environments, especially between March and November when tick activity is highest in both endemic and non-endemic areas, has been reported as the most likely cause of BB infection in forestry workers [46,54]. Being a member of this occupational category is not in itself a risk factor, the real risk lies in the type of tasks performed, and the operational setting Within the same category, performing manual activities in forests (e.g., woodcutting) was associated with higher BB seroprevalence than doing indoor administrative tasks [51,52,58,63,65,74]. The risk of contracting BB infection also increased if foresters ate meals in woodland while working outdoors [76], or performed recreational activities in natural environments, especially in gardens [58,59,60]. 

The frequency of BB infection was associated with different types of forest: deciduous and mixed-deciduous forests were more conducive to infection than coniferous forests [51,70]. 

Older age and a longer exposure to forest environments increased the likelihood of BB infection [51,55,58,59,60,61,63,65,66,67,70,76]. The frequency of bites was also associated with BB seroprevalence in foresters and farmers [43,44,45,48,49,51,58,59,60,61,63,64,67,68,69,73,74]. Seropositivity was significantly increased in persons reporting more than 1 [44] or 2 [67], or more than 5 [58,67,69] tick bites a year. 

Risk factors include awareness and application of safety measures. Poor knowledge of LD and low compliance with personal protective behaviors, were associated with higher rates of seropositivity [48,52,69]. The use of repellents, wearing clothes that prevent exposure of the skin, carefully checking the body for the presence of ticks after returning from the forest, showering after outdoor work, and the use of tweezers to rapidly remove ticks were actions associated with a decreased risk of BB infection [48,54,59,73]. 

### 3.5. Meta-Analysis

A total of 20 of the 35 studies included in the systematic review measured seroprevalence in exposed workers without including a control group and were therefore excluded from subsequent analyses. The meta-analysis therefore included 15 studies (3932 subjects). The estimated cumulative odds ratio was significantly higher in individuals who performed outdoor activities than in controls (OR 1.93; 95% CI 1.15; 3.23) (Table 2, Figure 2). 

The I^2^ index revealed a moderate level of heterogeneity between the studies (66.81%). Funnel plot showed some evidence of publication bias (Figure 3), mainly due to the absence of small negative studies. However, Egger’s test was non-significant (*p* = 0.712).

To interpret the heterogeneity observed, we analyzed the studies on forestry workers and farmers separately from the other categories. The estimated OR in forest and agricultural workers in comparison with controls was 2.36 (CI95% 1.28; 4.34), while there was a non-significant increase in risk (OR = 1.05, CI95% 0.28; 3.88) in the remaining categories of workers (veterinarians, animal breeders, and soldiers). (Table 3, Figure 4). 

We then checked whether the heterogeneity observed was related to the period in which the studies were conducted. In the studies published until 2010 we observed a significantly increased risk in exposed workers (OR 3.03 95%CI 1.39–6.61), while in more recent studies the odds became non-significant (OR 1.08 95% CI 0.63–1.85) (Table 4) (Figure 5).

## 4. Discussion

This study confirmed that LD is an occupational hazard for forestry and agricultural workers. A non-significant increase in risk may be observed in other categories of workers who enter infested areas (soldiers) or work with animals (veterinarians, breeders). Although we expected that climate change effects might contribute to LD infections by promoting the proliferation of ticks [77], we observed that the previously high occupational risk in outdoor workers had fallen over the past decade. The spread of prevention measures and knowledge of LD may have brought about this reduction in the occupational risk that now tends to overlap that of the general population.

Our systematic review identified frequent direct contact with ticks, or with animals that host ticks, as the main risk factors. The risk of being bitten by infected ticks increased directly with age, as well as with the time spent in the occupational category or outdoors in the forest. After the manufacture of Lyme vaccine was discontinued in 2002 [78], strategies to prevent contact with ticks and the duration of feeding have necessarily focused on the use of personal protective equipment (PPE) and the adoption of personal protective behaviors (PPB). These include using insect repellents on exposed skin, wearing permethrin-treated clothing while outdoors, conducting a full-body check, examining personal equipment, removing ticks with the proper technique, bathing or showering within two hours of exposure to remove ticks, and drying clothes at a high temperature setting for an hour in order to kill any remaining ticks [79,80,81,82,83]. Body examination after work and the prompt removal of ticks from skin with fingers, or more correctly with tweezers, are additional preventive measures that have been shown to help protect workers from infection transmission [83]. Since there is also a close relationship between adherence to preventive measures and knowledge of LD [81,83,84]. The education and training of outdoor workers play an essential role in prevention. The most probable explanation for the reduction in risk observed in recent studies could be the greater awareness of tick-transmitted diseases that has also led to workers’ adoption of the recommended PPB. Younger, well-trained forestry workers showed greater compliance with PPB and lower prevalence of LD than older workers [85]. Moreover, early diagnosis and treatment of LD have contributed to preventing seroconversion [48]. 

Piacentino and Schwartz (2002) [25] carried out the first systematic review of occupational LD by selecting 41 studies conducted between the 1980s and the end of the last century. They observed that the choice of an appropriate comparison group can influence the magnitude of this risk, and that a comparison of seroprevalence among occupationally exposed people living in LD-endemic areas with controls living in areas where LD is lowest or absent, would bias estimations of occupational risk in the direction of increased risk. They also observed that more recent studies documented no increase in the incidence of symptomatic, clinically confirmed LD in outdoor workers. A review of zoonoses in forestry workers [28] collected 22 studies published between 1995 and 2010, confirming not only the over-representation of positive seroprevalences for BB in forestry workers, but also the perplexities due to non-standardized methodologies, differences in the sample sizes of the populations studied, and the lack of control populations. A recent scoping review on zoonosis [31] cited 7 LD studies on forestry, farm workers and the military. In addition to supplementing these reviews with more recent studies, we aimed to assess the magnitude of the risk by conducting a meta-analysis. Our findings confirm the hypothesis of previous researchers, i.e., the risk for forestry workers is only modest, while the risk for veterinarians and soldiers is negligible and has shown a tendency to decrease in more recent years. 

Studies conducted in the workplace are of considerable importance for public health since they enable us to highlight the so-called natural foci where LD is endemic due to the environmental and ecological characteristics of those areas (e.g., presence of reservoir animals such as rodents, deer that host ticks, vegetation type). The spread of occupational LD parallels that in the general population where there are important regional differences. For example, in Europe, the estimated population-weighted average incidence rate for the regional burden of LB is 22.05 cases per 100,000 persons per year [86]. In Poland there were 45 cases per 100,000 persons in 2016, while in Lombardy in Northern Italy, there were only 1.24 new cases per 1 million residents between 2000 and 2015 [87]. Such marked differences must be an incentive for improving health literacy on this topic, and must prompt policymakers to strengthen preventive measures throughout the whole territory. 

Our review confirmed that many of the workers with positive antibodies never manifested symptoms. Latent infections are nevertheless of importance. Although asymptomatic infection generally has a good prognosis, and therefore do not require any measures, the same is not the case for neglected infections. Moreover, 1 out of 8 people who do not receive proper diagnosis and treatment develop pathological sequelae such as arthritis over the 12-month period following seroconversion [88,89]. The lack of symptomatology increases the under-reporting of disease because workers may not recognize the signs and symptoms of infections. To avoid any delays in diagnosis and treatment, thereby facilitating disease progression, occupational physicians play a strategic role by making sure regular serological tests are conducted on all workers at risk of BB infection, especially those employed in high endemic areas where the presence of a large quantity of infected ticks enhances the likelihood of transmission of infection after a bite. The adoption of a two-step procedure (ELISA or IFA followed by immunoblotting—Western blot—if there is a positive reaction) improves specific identification of infected workers. Workers should be informed about the importance of consulting their physician after a tick bite and, according to the clinical and serological assessment, consider receiving medical prophylaxis in accordance with clinical guidelines. 

This study has some limitations. The main weakness is the poor definition of occupational categories and exposure. With regards possible exposure to ticks, forestry workers and farmers in particular, do not carry out similar activities. Since none of the selected studies provided an exact definition of the type of exposure, individual cases of tick bites were often used as a proxy for exposure. Another limitation concerns the differing criteria used to define serological positivity. In the meta-analysis we included only studies that had a control group and used a similar method to evaluate seroprevalence. Although this drastically reduced the number of studies compared to the total included in the systematic review, there continued to be a high heterogeneity that we tried to analyze by stratification. The strength of this study lies in the fact that its analysis of the most recent literature provides useful indications for public health authorities regarding the effectiveness of current measures for preventing LD in the workplace. 

## 5. Conclusions

Occupational risk factors place many individuals at risk of tick bites and LD. This review showed that forester and farmers are the most exposed categories, and that occupational risk appears to have been reduced in the last decade, probably as a result of preventive measures. However, compared to the general population, there are differences in geographical exposure to ticks in work environments, especially among foresters and farmers. For this reason, strategies should be implemented to improve the awareness of the spread of LD among vulnerable workers. These could include combined action on the part of local public health agencies and labor/civil organizations to inform workers of the risks involved and the need to consult a physician after experiencing a tick bite.

## Figures and Tables

**Figure 1 diagnostics-12-00296-f001:**
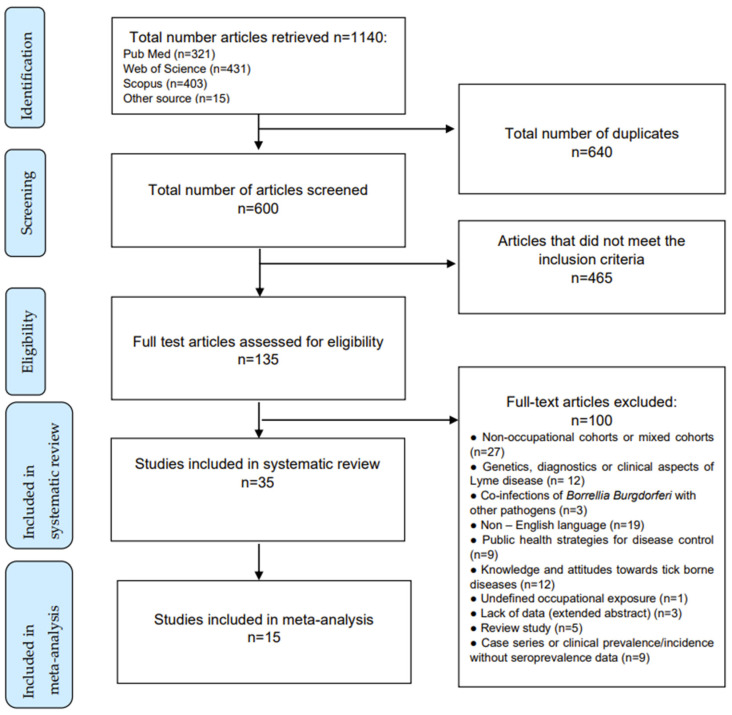
Article selection algorithm (PRISMA 2009).

**Figure 2 diagnostics-12-00296-f002:**
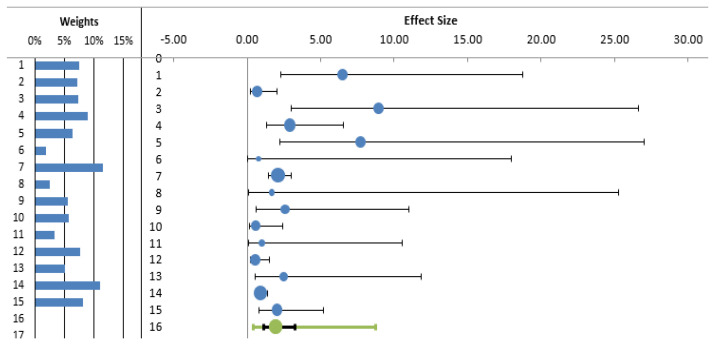
Forest plot of odds ratio for LD (diagnosed by antibodies) among exposed (outdoor) workers and other individuals not exposed (controls).

**Figure 3 diagnostics-12-00296-f003:**
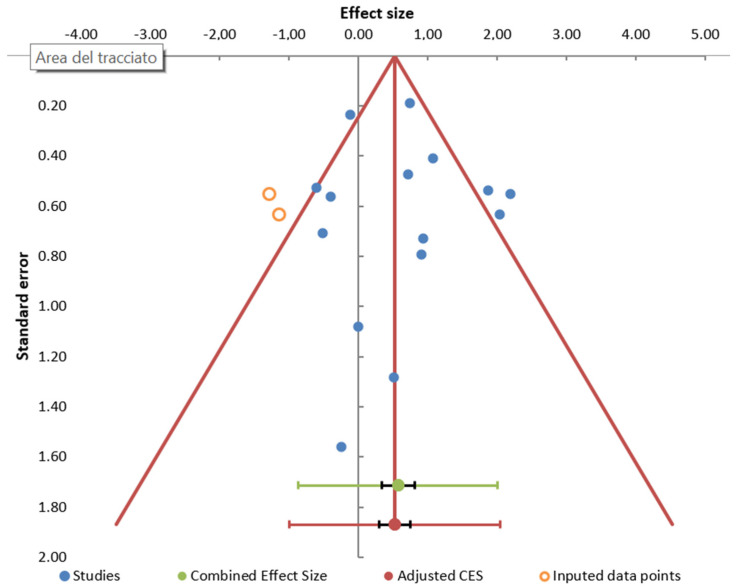
Publication bias. Funnel plot of studies reporting LD (diagnosed by antibodies) associated with occupational exposure.

**Figure 4 diagnostics-12-00296-f004:**
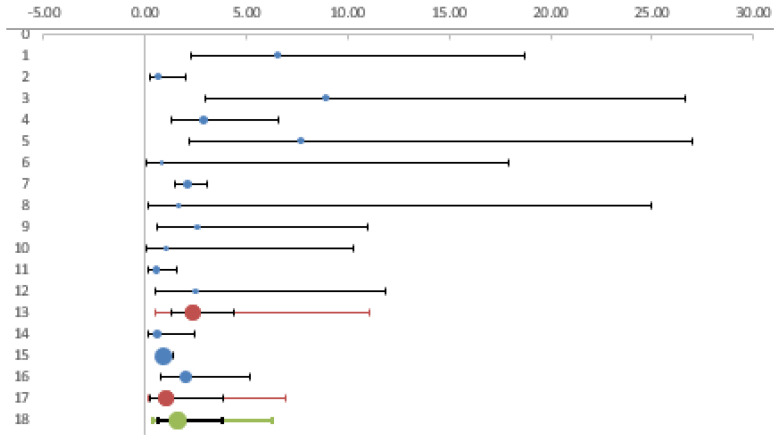
Forest plot. Foresters and farmers compared with other categories of workers.

**Figure 5 diagnostics-12-00296-f005:**
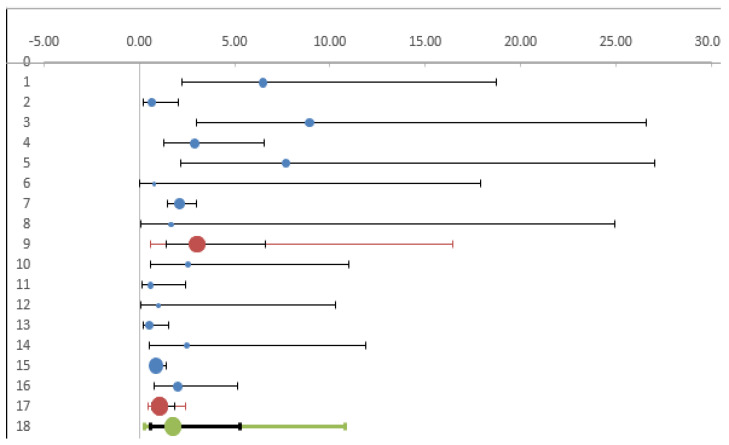
Forest plot. Comparison of studies published before and after 2011.

**Table 1 diagnostics-12-00296-t001:** Summary of the studies included in the systematic review (*n* = 35).

Author	Location	Type of Worker	Cases/Controls	Risk Factors	Diagnosis	Seroprevalence	Odds Ratio (Confidence Interval 95%)
Oehme et al., 2002 [42]	Baden-Wuerttemberg, Germany	Forestry	4368	None	ELISA, WB, CI	34.2% (WB)	-
Niscigorska et al., 2003 [43]	West Pomeranian, Poland	Forestry	52	Tick bites history	ELISA, CI	61.5% (ELISA)	-
Cinco et al., 2004 [44]	Friuli, Italy	Forestry	181	Tick bites history	ELISA, WB, CI	24.3% (ELISA); 23.2% (WB).	-
Santino et al., 2004 [45]	Central and Southern Italy	Outdoor	387/325	Tick bites history	ELISA; CI	7.5% (ELISA); 1.2% in controls.	7.09 (2.46–20.41)
Tomao et al., 2005 [46]	Tuscany, Italy	Outdoor	412/365	Tick bites history	ELISA, WB, CI	7.8% (ELISA), 3.9% (WB). 4.9% (ELISA), 1.6% (WB) in controls	2.41 (0.93–6.24)
Cisak et al., 2005 [47]	Southeastern Poland	Forestry	113/56	Tick bites history	ELISA, CI, PE	40.7% (ELISA) 7% in controls	8.9 (8.01–26.4),
Rojko et al., 2005 [48]	Slovenia	Forestry	122/93	Tick bites history.	IFA, ELISA	16.4% (IgM ELISA), 23.8% (IgG ELISA). 4.1% (IgM IFA), 9.8% (IgG IFA). 16.2% (IgM ELISA), 9.7% (IgG ELISA) 4.3% (IgM IFA), 4.3% (IgG IFA) in controls	2.90 (1.30–6.5)
Cisak et al., 2008 [49]	Lublin, Poland	Farmers	94/50	Tick bites history	ELISA, WB, CI	33% (WB). 6% in controls	7.71 (2.22–26.7)
Kaya et al., 2008 [50]	Duzce, Turkey	Forestry and Farmers	349/193	Tick bites history, animal contact	ELISA; IFA, WB, CI	10.9% (ELISA) 1.1% (WB) 2.6% (ELISA) 0% (WB) controls	4.6 (1.8–11.8)
Buczek et al., 2009 [51]	Southern Poland	Forestry	864/291	Age, male sex, length of occupation	ELISA	13.8% (IgM ELISA), 25% (IgG ELISA). 10% (IgM ELISA), 13,7% (IgG ELISA) IgG in controls	1.44 (1.00–2.08)
Di Rienzi et al., 2010 [52]	Latium, Italy	Forestry	145/282		ELISA, WB	3.4% (IgG ELISA), 13.1% (IgM ELISA) 0.69% (IgG WB), 6.2% (IgM WB). 3.2% (IgG ELISA), 7.1% (IgM ELISA) 1.06% (IgG WB), 6.73% (IgM WB) in controls.	1.08 (0.35–3.3)
Adjemian, 2012 [53]	USA	Park employees	141	Tick bites history, PPE, PPB	ELISA, WB	Absence of seroprevalence and seroconversion during follow-up	-
Cisak et al., 2012 [54]	Lublin, Poland	Forestry	82/14	Workers’ knowledge, PPE PPB	ELISA, PE	15.4% (IgM ELISA) 41.0% (IgG ELISA); 21.4% (IgG ELISA) in controls	2.5 (0.61–10.6)
Lakos et al., 2012 [55]	Hungary	Forestry	1670	Age, Male sex	ELISA, CI, PE	37% (ELISA)	-
Tabibi et al., 2013 [56]	Northern Italy	Animal breeders	64/32	None	IFA	7.8% (IFA IgG) 12.5% controls	0.59 (0.15–2.38)
Jovanovic et al., 2015 [57]	Serbia	Forestry,	69/35	None	ELISA, WB, CI	11.76% (IgG/M ELISA\WB), Controls 8.57%	1.42 (0.29–6.9)
Jurke et al., 2015 [58]	North Rhine, Germany	Forestry	722/228	Tick bites history, Age, Male sex, Outdoor work	ELISA, CI	30.6% (ELISA IgG)	3.96 (2.60–6.04)
Zákutná et al., 2015 [59]	Slovakia	Agricultural, forestry and police workers	277	Tick bites history, Age, Male sex, Activity, PPE, PPB, Contact with animal	ELISA, CI	25,3% (ELISA IgG) agricultural and foresters 29%; police officers 11%; outdoor workers 21%	
Bucak, 2016 [60]	Turkey	Agricultural workers	196/113	Tick exposure history, age, Female sex, Low education, habitat	ELISA, WB	10.8% (IgG/IgM WB) 6.2% controls	1.84 (0.65–5.16)
De Keukeleire et al., 2016 [61]	Belgium	Farmers, veterinarians	31/96	Age, Male sex, habitat	ELISA, CI	9.68% (ELISA IgG), 4.17% veterinarians	2.46 (0.51–167)
Gazi et al., 2016 [62]	Manisa, Turkey	Farmers, hunters	324	Younger age (<50 years)	IFA, WB	0.9% (WB IgG)	0.66 (0.05–7.49)
Rigaud et al., 2016 [63]	Northeastern France	Forestry	2975	Age, manual task, habitat	ELISA; WB, CI	14.1% (IgG WB)	2.22 (1.29–3.82)
Skinner-Taylor M et al., 2016 [64]	Mexico	veterinarians	40	Tick bites history	ELISA, WB, CI	47.5% (ELISA IgG), 22.5% (IgG WB)	-
Cora et al., 2017 [65]	Trabzon, Turkey	Farmers, military	555/329	Age, Male sex, Low Education level	ELISA, WB	28.1% (IgG ELISA), 15.9% (IgG WB); 24.3% (IgG ELISA), 12.2% (IgG WB) controls	1.36 (0.91–2.03)
De Keukeleire et al., 2017 [66]	Belgium	Veterinarians, farmers	148/402	Age	ELISA, CI	5.4%, (IgG ELISA), 2.7% (IgG ELISA) controls	2.03 (0.8–5.15)
Zajac et al., 2017 [67]	Poland	farmers	3597	Tick bites, Age, residence	ELISA, CI, PE	11.5% (IgM ELISA) 13.7% (IgG ELISA)	-
Bušová et al., 2018 [68]	Slovakia	Gardeners, soldiers	135/126	Tick bites history	ELISA, CI	15.6% (IgG ELISA) 5.9% (IgM ELISA). 19% IgG ELISA 5.6% (IgM ELISA) controls	-
De Keukeleire et al., 2018 [69]	Belgium	Forestry	310	Tick bites history, PPB, forest work	ELISA, CI	21.6% (IgG ELISA).	-
Kiewra et al., 2018 [70]	Poland	Forestry	646	Age, forest work	ELISA, WB	22% (IgM/IgG WB) 19% (IgM ELISA) 8.7% (IgM WB) 29.1% (IgG ELISA), 17.8% (IgG WB).	-
Van Beek 2018 [71]	Finland	Forestry and farmers	24	None	ELISA, WB	4.1% (IgG WB)	3.59 (0.48–22.20)
Lledó et al., 2019 [72]	Spain	Forestry	100	None	IFA, CI	7% (IgG IFA)	-
Pańczuk et al., 2019 [73]	Lubin, Poland	Forestry and farmers, hunters	150	Tick bites history, PPB	ELISA, WB, CI	10% (IgM ELISA) 53.3% (IgG ELISA), 0.6% (IgM WB), 26% (IgG WB)	-
Babu et al., 2020 [74]	India	Forestry	518	Tick bites history, manual task	ELISA; WB, CI	19.9% (ELISA), 3% (WB)	-
Cuellar et al., 2020 [75]	Finland	Forestry and farmers	283	None	ELISA; WB	20.8% (IgG WB)	-
Acharya and Park, 2021 [76]	South Korea	Park employees	655	Age, outdoor work	IFA, CI	8.1% (IgM/IgG IFA)	-

Notes: ELISA enzyme-linked immunosorbent assay; IFA immunofluorescent assay, WB western blot; CI clinical interview; PE physician’s examination; PPB, personal protective behaviour; PPE, personal protective equipment.

**Table 2 diagnostics-12-00296-t002:** Comparison of outdoor workers with controls. Meta-analysis of 15 studies.

Study	OR	CI Lower Limit	CI Upper Limit	Weight
1. Santino et al., 2004 [45]	6.50	2.26	18.73	7.46%
2. Tomao et al., 2005 [46]	0.67	0.22	2.05	7.20%
3. Cisak et al., 2005 [47]	8.93	3.00	26.60	7.30%
4. Rojko et al., 2005 [48]	2.91	1.30	6.53	9.96%
5. Cisak et al., 2008 [49]	7.71	2.20	27.03	6.46%
6. Kaya et al., 2008 [50]	0.78	0.03	17.93	1.89%
7. Buczek et al., 2009 [51]	2.09	1.45	3.02	11.49%
8. Di Rienzi et al., 2010 [52]	1.67	0.11	25.26	2.60%
9. Cisak et al., 2012 [54]	2.55	0.59	11.00	5.61%
10. Tabibi et al., 2013 [56]	0.59	0.15	2.42	5.77%
11. Jovanovic, 2015 [57]	1.00	0.10	10.52	3.38%
12. Bucak, 2016 [60]	0.54	0.19	1.53	7.60%
13. De Keukeleire et al., 2016 [61]	2.46	0.51	11.85	5.08%
14. Cora et al., 2017 [65]	0.89	0.56	1.41	11.02%
15. De Keukeleire et al., 2017 [66]	2.03	0.80	5.16	8.18%
Combined effect size	1.93	1.15	3.23	
Heterogeneity-I^2^	66.81%			

OR = odds ratio; CI: confidence interval at 95%.

**Table 3 diagnostics-12-00296-t003:** Comparison of Forester and farmers with other categories of exposed workers.

Study	OR	CI Lower Limit	CI Upper Limit	Weight
Santino et al., 2004 [45]	6.50	2.26	18.73	9.94%
Tomao et al., 2005 [46]	0.67	0.22	2.04	9.59%
Cisak et al., 2005 [47]	8.93	3.00	26.60	9.73%
Rojko et al., 2005 [48]	2.91	1.30	6.53	12.01%
Cisak et al., 2008 [49]	7.71	2.20	27.03	8.58%
Kaya et al., 2008 [50]	0.78	0.03	17.93	2.47%
Buczek et al., 2009 [51]	2.09	1.45	3.02	15.54%
Di Rienzi et al., 2010 [52]	1.67	0.11	24.94	3.41%
Cisak et al., 2012 [54]	2.55	0.59	11.00	7.43%
Jovanovic, 2015 [57]	1.00	0.10	10.31	4.44%
Bucak, 2016 [60]	0.54	0.19	1.53	10.13%
De Keukeleire et al., 2016 [61]	2.46	0.51	11.85	6.72%
Foresters and Farmers, combined	2.36	1.28	4.34	51.22%
Tabibi et al., 2013 [56]	0.59	0.15	2.42	14.91%
Cora et al., 2017 [65]	0.89	0.56	1.41	57.55%
De Keukeleire et al., 2017 [66]	2.03	0.80	5.16	27.55%
Other categories, combined	1.05	0.28	3.88	48.78%
Combined effect size	1.59	0.67	3.78	

**Table 4 diagnostics-12-00296-t004:** Comparison of studies performed before and after 2011.

Study	OR	CI Lower Limit	CI Upper Limit	Weight
Santino et al., 2004 [45]	6.50	2.26	18.73	13.92%
Tomao et al., 2005 [46]	0.67	0.22	2.04	13.43%
Cisak et al., 2005 [47]	8.93	3.00	26.60	13.62%
Rojko et al., 2005 [48]	2.91	1.30	6.53	16.90%
Cisak et al., 2008 [49]	7.71	2.20	27.03	11.98%
Kaya et al., 2008 [50]	0.78	0.03	17.93	3.41%
Buczek et al., 2009 [51]	2.09	1.45	3.02	22.93%
Di Rienzi et al., 2010 [52]	1.67	0.11	24.94	4.71%
Until 2010	3.03	1.39	6.61	47.14%
Cisak et al., 2012 [54]	2.55	0.59	11.00	8.16%
Tabibi et al., 2013 [56]	0.59	0.15	2.42	8.56%
Jovanovic, 2015 [57]	1.00	0.10	10.31	3.93%
Bucak, 2016 [60]	0.54	0.19	1.53	14.26%
De Keukeleire et al., 2016 [61]	2.46	0.51	11.85	6.98%
Cora et al., 2017 [65]	0.89	0.56	1.41	41.29%
De Keukeleire et al., 2017 [66]	2.03	0.80	5.16	16.81%
After 2011	1.08	0.63	1.85	52.86%
Combined effect size	1.76	0.58	5.29	

## Data Availability

The review protocol was deposited on 2 May 2021, in the international prospective registry for systematic review protocols PROSPERO (registration number: CRD42021252608). Data used for meta-analysis are available in Zenodo, 10.5281/zenodo.5894819.

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
