# Peer review of "Occupational Lyme Disease: A Systematic Review and Meta-Analysis"

_diagnostics, 2022, doi:10.3390/diagnostics12020296_

Round 1
Reviewer 1 Report
Review
Occupational Lyme Disease: A Systematic Review and 2 Meta-Analysis
Nicola Magnavita, Ilaria Capitanelli 3, Olayinka Ilesanmi 4 and Francesco Chirico
This systematic review focus on Lyme Disease related to occupation activities and meta-analysis, including updated reports. Environmental changes and the consequences in emerging health problems, including Lyme Disease, are high interest topics in the whole world.
Comments
- The conception and methodology are appropriate for the subject and the article type, according PRISMA (systematic reviews and meta-analyses).
- The manuscript is well structured.
- Figures and tables are appropriate and sensitive.
- Considering the title of the Journal Diagnostics, it should be added in the Introduction section a short review on serologic diagnostic of Lyme Diseases, the diversity of Borrelia species, the influence on the results with ELISA tests, the sensitivity of ELISA tests.
Author Response
Reviewer #1
This systematic review focus on Lyme Disease related to occupation activities and meta-analysis, including updated reports. Environmental changes and the consequences in emerging health problems, including Lyme Disease, are high interest topics in the whole world.
Comments
The conception and methodology are appropriate for the subject and the article type, according PRISMA (systematic reviews and meta-analyses).
The manuscript is well structured.
Figures and tables are appropriate and sensitive.
Considering the title of the Journal Diagnostics, it should be added in the Introduction section a short review on serologic diagnostic of Lyme Diseases, the diversity of Borrelia species, the influence on the results with ELISA tests, the sensitivity of ELISA tests.
Response: we thank the reviewer very much for the appreciation of our work. We fully agree on the opportunity to supplement the Introduction with a short note on the diagnostic significance of physiological diagnoses in Lyme disease. We have changed the text, which is now as follows:
“…LD diagnosis is challenging due to the varied clinical manifestations it may present [6] and is supported by serologic testing using a 2-step process [13, 14]. Current recommendations based on the 1995 guidelines of the Centers for Disease Control and Prevention include using a sensitive enzyme immunoassay or immunofluorescence assay, such as enzyme-linked immunosorbent assay (ELISA) or immunofluorescent assay (IFA), followed by a western immunoblot assay for specimens yielding positive or equivocal results [15]. In 2019, the Food and Drug Administration (FDA) cleared several LD serologic assays with new indications for use, allowing for enzyme immunoassay than western immunoblot assay as the second test in a Lyme disease testing algorithm [16]. The two-step approach is intentionally conservative to exclude false positive results; consequently, it is prone to false negative results that lead to underestimation of the number of people with LD. In Canada, a 10 to 20-fold underestimate has been calculated [17]. The Western blot /immunoblot assay confirms a percentage between 30% and 70% of positivity at first assay [18, 19] However, some epidemiological studies, especially in past years, have used a single ELISA test approach for detecting BB infection.

Reviewer 2 Report
Well written manuscript with interesting scope on summarizing data on occupational Lyme disease/seroprevalence., significance of results are likely to be limited
-lines 43-46: the authors emphasize suicidality, perhaps it would be better to discuss mental diseases/disorders in general as the sentence gives the impression that suicidality is a very important sequela of Lyme disease, but the manuscript does not discuss this later.
-lines 52-53: references needed and the relevance of the issue of ELISA screening without confirmation for the results of studies should be mentioned.
-Line 75-78: the sentence suggests as if PROSPERO had reviewed the protocol (PROSPERO does not assess the quality of submitted protocols). It would be better to use "the protocol was registered..." or similar.
-lines 321-335: the paragraph suggests that asymptomatic workers who seroconvert should be treated. This cannot be deducted from the results in the manuscript. Not being aware of such recommendations, the authors need to cite guidelines or published evidence if they considers this to be true. On the same line the guidelines mentioned by the authors suggesting receiving medical prophylaxis after tick bites (lines 333-335) needs to be cited.
Author Response
Reviewer #2
Well written manuscript with interesting scope on summarizing data on occupational Lyme disease/seroprevalence., significance of results are likely to be limited
-lines 43-46: the authors emphasize suicidality, perhaps it would be better to discuss mental diseases/disorders in general as the sentence gives the impression that suicidality is a very important sequela of Lyme disease, but the manuscript does not discuss this later.
Response: This observation gives us the welcome opportunity to modify the text. Now we have written: …
“…These symptoms include a bullseye-shaped rash (Erythema migrans), fever, chills, headache, fatigue, muscle and joint pain, and swollen lymph nodes [3,4]. If LD is diag-nosed and treated with antibiotics, most cases do not go beyond a skin rash. However, if it is left undetected, various long-lasting neurologic, cardiovascular, and musculo-skeletal long-term consequences may occur [5]. The late disseminated stage manifests with acrodermatitis atrophicans, Lyme arthritis, and neurological symptoms can be se-riously debilitating [6]. The immune-mediated and metabolic changes can induce mul-tiple debilitating symptoms and alter neural circuits. Neurological manifestations, termed Lyme neuroborreliosis, occur in about 10% of patients with LD [7, 8]. A subset of patients may report persistent symptoms, including severe fatigue, anxiety and de-pression [9]. A small percentage (~10%) of patients may go on to develop a poorly de-fined fibromyalgia-like illness, post-treatment Lyme disease symptoms [10], whose characterization is still controversial. Hospitalized patients have an increased risk of mental disorders, affective disorders, suicide attempts and suicide deaths compared with other patients without LD [11]. Results of indirect estimates reveal that there may be more than 1,200 LD-associated suicides in the US per year [12].”
-lines 52-53: references needed and the relevance of the issue of ELISA screening without confirmation for the results of studies should be mentioned.
Response: Also accepting the request of another reviewer, we have defined in greater detail the diagnostic significance of serological tests, including the most recent indications from the FDA and the CDC. We have corrected the quoted sentence. The text now is as follows:
…LD diagnosis is challenging due to the varied clinical manifestations it may present [6] and is supported by serologic testing using a 2-step process [13, 14]. Current recommendations based on the 1995 guidelines of the Centers for Disease Control and Prevention include using a sensitive enzyme immunoassay or immunofluorescence assay, such as enzyme-linked immunosorbent assay (ELISA) or immunofluorescent assay (IFA), followed by a western immunoblot assay for specimens yielding positive or equivocal results [15]. In 2019, the Food and Drug Administration (FDA) cleared several LD serologic assays with new indications for use, allowing for enzyme immunoassay than western immunoblot assay as the second test in a Lyme disease testing algorithm [16]. The two-step approach is intentionally conservative to exclude false positive results; consequently, it is prone to false negative results that lead to underestimation of the number of people with LD. In Canada, a 10 to 20-fold underestimate has been calculated [17]. The Western blot /immunoblot assay confirms a percentage between 30% and 70% of positivity at first assay [18, 19] However, some epidemiological studies, especially in past years, have used a single ELISA test approach for detecting BB infection.
-Line 75-78: the sentence suggests as if PROSPERO had reviewed the protocol (PROSPERO does not assess the quality of submitted protocols). It would be better to use "the protocol was registered..." or similar.
R.: it's right. We have corrected the text, specifying that the protocol was registered (not “reviewed”) on PROSPERO with number CRD42021252608
-lines 321-335: the paragraph suggests that asymptomatic workers who seroconvert should be treated. This cannot be deducted from the results in the manuscript. Not being aware of such recommendations, the authors need to cite guidelines or published evidence if they considers this to be true. On the same line the guidelines mentioned by the authors suggesting receiving medical prophylaxis after tick bites (lines 333-335) needs to be cited.
R.: We completed the first sentence, which being incomplete led to a wrong interpretation. The second sentence was also incomplete, and this too could have caused misunderstanding. We thank the reviewer for pointing out two critical points that could have led to misunderstandings.

Reviewer 3 Report
The conclusions should be an answer to the aims of the study and result from the research presented in the manuscript, so the Authors should adjust them them to the chosen study aims.
Author Response
Reviewer3
The conclusions should be an answer to the aims of the study and result from the research presented in the manuscript, so the Authors should adjust them them to the chosen study aims.
R.: We thank the reviewer for this timely observation. We added the concept that: "This review showed that forester and farmers are the professionally most exposed categories and that occupational risk appears to be reduced in the last decade, probably as a result of preventative measures.”
